# Contribution of own food production to household dietary diversity and nutrient intake adequacy among refugee households in Palorinya, northwestern Uganda

**Martine Asobasi**[1,2]*, **Saidi Appeli**[3], **Peter Omara**[4], **Solomon Olum**[2]

**1** Department of Science, Faculty of Biology, Muni University, Arua City, Uganda, **2** Department of Food Science and Postharvest Technology, Faculty of Agriculture and Environment, Gulu University, Gulu City, Uganda, **3** Department of Agribusiness and Extension, Faculty of Agriculture and Animal Sciences, Busitema University, Soroti, Uganda, **4** Department of Soil and Crop Sciences, Texas A&M University System, College Station, Texas, United States of America

* asobasimartine@gmail.com

## Abstract

### Background

Hunger and malnutrition are major global issues particularly in conflict and disaster-affected areas. Own food production has been considered as effective strategy to improve nutrition outcomes. However, evidence on its contribution to dietary diversity and nutrient intake adequacy, especially in refugee settings with limited access to land remains limited. This study assessed the contribution of own food production to dietary diversity and nutrient intake adequacy among refugees in Palorinya settlement, northwestern Uganda.

### Methods

A cross-sectional survey was conducted among 316 randomly selected households in Palorinya refugee settlement, northwestern Uganda. Current household dietary diversity score (HDDS) was assessed using a 24-hour dietary recall of food groups consumed. Nutrient intake adequacy was estimated using food composition tables based on annual food production and consumption. Poisson regression with robust standard errors and multiple linear regression were employed to identify factors associated with current HDDS and nutrient intake adequacy. Beta coefficients with their 95% confidence intervals (CIs) were presented.

### Results

Own food production greatly contributed to vegetable consumption (85.0%) but was low for animal-sourced foods (3% for milk/dairy products; 7.5% for meat). Own food production contributed minimally to nutrient intake adequacy, with most households

**Data availability statement:** All relevant data are within the manuscript and its Supporting Information files.

**Funding:** The author(s) received no specific funding for this work.

**Competing interests:** The authors have declared that no competing interests exist.

falling short of annual recommended dietary allowance (RDA) for calcium (100%), iron (95%), zinc and protein (95.2%), and energy (99%). Education level, access to agricultural land, and kitchen garden were key predictors of both HDDS and nutrient intake adequacy. Additionally, size of agricultural, household income and occupation also influenced HDDS and nutrient intake adequacy.

## Conclusion

Own food production contributes moderately to dietary diversity and minimally to nutrient intake adequacy. Education level, land access, and kitchen gardening play important roles in shaping HDDS and nutrient intake adequacy. Interventions should focus on promoting kitchen gardening, nutrition-sensitive agriculture, and improving refugee land policy to enhance nutrient intake in refugee settlements.

## Introduction

Global efforts to end hunger and malnutrition have increasingly shifted from focusing on food quantity to enhancing dietary quality and nutritional adequacy. The shift to dietary quality are explicitly recognized in the United Nations (UN) Sustainable Development Goals (SDGs), specifically SDG 2 and SDG 3 which aim to eliminate hunger and promote health worldwide by 2030 [1]. Despite progress in food production, access to safe, diverse and nutritious food threatens the achievement of SDG 2. Consequently, hunger and malnutrition remain a major global challenge, with an estimated 720 million people experiencing chronic hunger in 2024 and around 2.6 billion unable to afford a healthy diet [2]. Sub-Saharan Africa (SSA) with an estimated 280 million undernourished people bears a disproportionate burden, where climate variability, food price shocks and conflict continue to undermine food and nutrition security [2–4]. Within this context, agriculture is now viewed not only as a source of livelihood and food but also as a pathway to improved nutrition through nutrition-sensitive agriculture (NSA)- an approach that intentionally integrates nutrition objectives into agricultural systems [5,6].

A core principle of NSA is own food production, where households produce crops or livestock primarily for their own consumption [7], thereby influencing food access, diet composition, and nutrient intake. This aligns with the food systems approach, which conceptualizes nutrition outcomes as functions of interactions between food production, access, consumption, and utilization [8].The production–consumption pathway, as described by Sibhatu and Qaim [9], posits that households consuming a larger share of own produced food tend to have higher dietary diversity and better micronutrient adequacy, especially where markets are underdeveloped or food prices are unstable. Consequently, own food production can enhance food and nutrition security resilience and self-sufficiency, particularly in fragile and conflict-affected settings. Empirical evidence from low- and middle-income countries (LMICs) supports the positive linkage between own food production, dietary diversity and nutrient intake adequacy. Studies across Africa, Asia, and Latin America show that each

additional crop or livestock species produced on-farm increases household dietary diversity scores (HDDS) and micronutrient adequacy ratios (MARs) [10–12]. In Uganda, Marivoet and Ulimwengu, [13] demonstrated that households that consumed food from their own diversified agricultural systems achieved up to 25% higher nutrient adequacy across energy, protein, and micronutrients compared to those relying solely on markets for their food consumption. Similarly, in Malawi, Luckett et al. [14] reported that households that consumed food from own food production through crop diversification had improved intake of iron, zinc, and vitamin A. These findings underscore the nutritional potential of own food production, particularly in contexts where markets are unreliable.

However, evidence also suggests that own food production does not automatically translate into improved nutrition outcomes [11,15]. For instance, in eastern and northern Uganda, despite participation in agriculture, many households still exhibit low dietary diversity and nutrient intake adequacy, primarily consuming cereals, legumes, and roots with limited animal-sourced or fruit-based foods [16]. Similarly, insufficient nutrient intake adequacy [17] has been found among households consuming food from own food production. The strength of this pathway depends on complementary factors such as women's empowerment, nutrition knowledge, and access to inputs [18,19]). Literature also points to other factors associated with household dietary diversity and nutrient intake adequacy. For instance, improved income status has been shown to influence dietary diversity [20,21]. Other factors include family size [22], gender [20], land ownership [23], and education level [24].

Uganda currently hosts over 1.6 million refugees, primarily from South Sudan and the Democratic Republic of Congo, making it the largest refugee-hosting country in Africa [25] Under the Self-Reliance Strategy (SRS), the Ugandan government allocates small land plots to refugee households to encourage own food production and reduce dependency on food aid. However, unpredictable rainfall and, often limited barren land plots limit refugees' ability to cultivate diverse nutrient rich crops and rear livestock [26,27]. As such, in most of the refugee settlements in Uganda, households frequently depend on food rations that are cereal-heavy and lack dietary diversity, exposing them to macro- and micronutrient deficiencies. For example, the Palorinya refugee settlement in northwestern Uganda has an 11% rate of global acute malnutrition (GAM) [28], a situation described as "serious" by the World Health Organization (WHO). Yet macro and micronutrient deficiencies leads to impaired cognitive growth and development, increased risk of morbidity and mortality, lower resistance to diseases, reduced physical and economic productivity, and stunting in children [29,30]. Against these detrimental effects, dietary diversity, measured by the number of food groups consumed within a reference period is widely recognized as a proxy for dietary quality and nutrient [15,31,32] Similarly, nutrient intake adequacy, the extent to which individual or household nutrient intakes meet the Recommended Dietary Allowance (RDA) for energy, macronutrients, and micronutrients [33], are critical aspect of the broader concept of food and nutrition security and continues to be advocated for averting hidden hunger [15], and improving nutritional status essential for human development [34]. Intake of adequate nutrients and diverse food groups is associated with improved nutrition outcomes. For example, Moursi et al. [32] asserts that increased dietary diversity increases micronutrient adequacy which improves health and nutrition outcomes in children and women anthropometry. Also, among women of reproductive age, consumption of diverse adequate nutrients is associated with improvements in health and nutrition status for both the mother and the fetus [29,30].

Nevertheless, despite the existence of intervention programs that promote refugee household food production in Palorinya refugee settlements, such as a portion of small pieces of land, few studies have quantified how own food production contributes simultaneously to both dietary diversity and nutrient intake adequacy in refugee contexts, where food systems are severely disrupted and production opportunities are limited primarily by land scarcity. The intersection of agriculture, displacement, and nutrition thus presents a critical research gap which limits the design of effective nutrition-sensitive interventions and agricultural support programs in refugee settlements. Therefore, this study examines the contribution of own food production to current household dietary diversity and nutrient intake adequacy among refugee households in the Palorinya settlement of northwestern Uganda. Specifically, it assesses: (1) the extent to which own food production contributes to the current household dietary diversity; (2) the degree to which own food production meets nutrient intake

adequacy for energy, protein, and key micronutrients; and (3) the socioeconomic factors influencing these outcomes. By integrating the concepts of nutrition-sensitive agriculture and food systems resilience, this study provides empirical insights that can inform policy reforms on refugee self-reliance, sustainable land use, and nutrition improvement strategies, particularly in light of declining food aid, increasing climatic shocks, and the growing recognition that sustainable food systems must integrate agricultural, nutritional, and social resilience dimensions. The findings aim to guide humanitarian and development stakeholders in designing context-specific interventions that enhance dietary quality and promote sustainable nutrition security among displaced populations.

## Materials and methods

### Study design and setting

This study was a cross-sectional research design conducted in the Palorinya refugee settlement, located in Obongi District, Uganda. Palorinya is the second-largest refugee settlement in Uganda, with an estimated population of 129,513 refugees. Of this population, 53% are females, and 25% are youths, living in 26,922 households. The settlement is divided into seven zones: Morobi, Dongo West, Chinyi, Basecamp, Ibakwe, Dongo East, Belameling, and Budri. The primary economic activity in the settlement is small-scale farming. Palorinya was chosen as the study area due to its high Global Acute Malnutrition (GAM) rate of 11% [28], which is classified as "serious" by the World Health Organization (WHO). Additionally, the small plots of land allocated to refugees are often unsuitable for cultivation due to declining soil fertility caused by continuous farming further justifying the choice of this area for the study [16].

### Study population and sampling

The study population consisted of refugee households in Palorinya. The primary participants were caregivers, defined as adults responsible for food preparation. In rural settings like Palorinya, caregivers are typically women aged 18–59 years [35]. Caregivers were selected because they are involved in both food production and preparation [35]. We included caregivers responsible for food preparation the previous day and who agreed to participate in the study whereas caregivers who were not responsible for food preparation the previous day were excluded. Sample size estimation followed Fisher and Hall's formula described below.

$$n = \frac{z^2 pq\ deff}{d^2}$$

where;

  n = required sample size
  z = standard normal deviation (1.96 at 95% confidence level)
  p = proportion of the target population with the problem (11% GAM prevalence) [28]
  q = 1 - p (0.89)
  deff = design effect for cluster sampling (conservatively set at 2 to account for potential biases)
  d = degree of precision (0.05)
  Including a 5% attrition rate, the calculated sample size was 315 households. However, 316 households were included in the final analysis to ensure consistency and completeness of data.

  A multistage sampling approach was used to select participants. In the first stage, the Palorinya refugee settlement was purposively selected on the basis of high malnutrition rates. In the second stage, five zones (Morobi, Dongo West, Chinyi, Belameling, and Budri) were randomly selected. In the third stage, 63 households from each zone were systematically selected. Here, we calculated the sampling interval by dividing the population by the sample size. The starting household was chosen randomly, and subsequent households were selected at fixed intervals. Finally, in households with more than

one caregiver, the one responsible for food purchase, preparation, and allocation was selected for the interview. A structured questionnaire was developed using an open data kit (ODK) and administered through face-to-face interviews for about 45–60 minutes, to collect quantitative data. Data were collected from July 20, 2024 to August 10, 2024. This data collection period aligns with the first harvest season in northwestern Uganda, and there was no food distribution which in the refugee settlement which would otherwise have affected dietary diversity scores. The questionnaire had three sections: Section 1 collected socio-economic data (e.g., household size, education level, occupation, marital status, and livelihood sources). Section 2 assessed current household dietary diversity using a 24-hour recall method based on 12 food groups recommended by Food and Agricultural Organization of the United Nations (FAO) [36], and section 3 captured data on annual household food production and consumption.

### Quality control measures

The questionnaire was pre-tested on 10 respondents in Dongo East (not part of the study) who were interviewed twice within seven days to ensure consistency, validity and reliability. Six research assistants (Nutritionists) and six Village Health Teams (VHTs) underwent training on the study protocol before conducting the interviews. During the pretesting, the research assistants established local units of measure commonly used in the refugee settlements and estimated the standard units of the local measures in either full or half full for commonly produced and consumed foods in the settlement. Data were stored in an encrypted master file to prevent unauthorized access, daily feedback sessions were held to address errors in data collection, and data was entered twice to ensure accuracy.

### Variables and measurements

The outcome variables were current household dietary diversity score (HDDS), and nutrient intake adequacy. The HDDS was assessed using the 24-hour recall method to determine the consumption of different food groups from own food production. Instead of using the 12 food groups proposed by FAO, nine food groups were used. Previous studies have used HDDS to assess nutrition quality [37,38]The food groups included (milk and milk products, eggs, fish, meat, fruits, vegetables, roots and tubes, pulse/nuts cereals and grains). Sugar/sweets, fats and oils were not included as they contribute a low amount of nutrients [9]. past studies involving nutrition quality assessment also used nine food groups to calculate HDDS. [12,39]. In the household dietary diversity questionnaire, respondents were asked whether they consumed or not consumed each of the food groups above. For each food group consumed, a score of 1 was given while for each food group not consumed, a score of 0 was given [36]. Scores from all the nine food groups were added to measure household dietary diversity as a count outcome. To assess nutrient intake adequacy, data were collected on estimates of annual crop food production and consumption of the households using sacks, basins, tins, and all the quantities converted into standard unit of kilograms and grams [40].The energy and nutrients contributed by each food was estimated by utilizing food composition table for eastern and central Uganda developed by harvest plus, which categorizes nutrient composition per 100g of edible portion [41]. In the event of missing values, East Africa food composition table was conducted and all energy and selected nutrients expressed in annual terms. The RDA of each household was calculated by utilizing FAO & WHO [42] [energy and dietary reference cut-off points, taking into consideration family size and composition [13]. Based on actual and recommended total nutrient consumption, Nutrient Household Adequacy (NHA), which measures nutrient and energy distribution in each households was obtained by dividing total energy and nutrient consumed from own food production by recommended intake [13]. Households were categorized as adequate when values were above one while households with values below one were considered to have inadequate intake before multiplying all by 100 [10].

The independent variables included age (categorized as < 18, 18–30, 31–45, and 46 and above), gender of the household head (male or female), marital status (married, separated, divorced, single, widowed), educational level (none, primary, secondary and tertiary), occupation (unemployed, employed, small scale trading, casual labour, farming).unemployed refers to household heads not engaged in any formal or informal income generating activity at the time of the

survey, including farming, trading or casual labour. Additional independent variables include household size (small(1–4 members), medium (5–10 members), and large(>10 members), religion (catholic, Muslim, protestant, seventh day Adventist, other), household participation in agriculture (yes vs. no), access to agricultural land (yes or no), acres of agricultural land (less than1 acre vs more than 1 acre), presence of kitchen garden (yes vs no), source of income (sale of crops, sale of animals, casual labour, brewing alcohol, small scale business, hand craft, and petty trade), source of food (own food production, market purchase, gifts/donations), distance to market (less than 1 km vs more than 1 km), access to hired land (yes vs no), monthly income and expenditure in Uganda shillings.

## Statistical analysis

Data management and analysis were performed via Stata/SE version 15.1. We descriptively summarized categorical data using frequencies and percentages and numerical data using means and standard deviations if normally distributed while median and interquartile ranges were used when skewed. To examine the predictors of HDDS, we fit a Poisson regression model for count data with a log link, and robust standard errors to identify the factors independently associated with current household dietary diversity. The independent variables included in the Poisson model were specified *a priori*, including variables supported by prior literature as relevant predictors of the outcome. Statistical significance was set at $p < 0.05$. To minimize the effect of multi-collinearity, independent variables with a variance inflation factor (VIF) of greater than 5 were considered multi-collinear and were excluded from the model. We conducted a link specification test which involves adding the square of the estimated linear predictors as an extra independent variable and assessing the fall in deviance to assess the appropriateness of the log function. A non-significant result indicates correct link specification. Additionally, we tested the Poisson assumption that the variance is equal to the mean using the Pearson-based dispersion statistic with one degree of freedom. The assumption holds when the Pearson-based dispersion statistic value is not greater than one [43]. To assess the predictors of nutrient intake adequacy, we built six empirical multiple linear regression models as a function of the independent variables selected in the Poisson model to independently identify factors that predict adequate nutrient intake for the selected nutrients. Model 1 was a multiple linear regression model for energy, model 2 for protein, model 3 for calcium, model 4 for iron, model 5 for zinc, and model 6 for vitamin A. To account for data aspects such as non-constant variance or outliers, we used a weighted least squares in each of the multiple linear regression models.

## Ethical statement

This study was approved by the Gulu University Research and Ethics Committee (GUREC-2023–574, dated August 17, 2023). Written informed consent was obtained from the parents or guardians responsible for food production and preparation after receiving detailed explanations about the study procedures, potential benefits, risks, and voluntary participation, including the right to opt out of the study at any stage. Individuals who did not participate in the study did not receive any treatment similar to those that participated. Confidentiality was maintained through anonymized data collection and secure storage.

## Results

### Characteristics of the participants

Of the 316 participants studied (Table 1), the majority 212 (67.1%) were male, aged between 32–46 (46.2%) years, and married 236 (74.7%) at the time of the survey. About half, 157 (49.7%) were not employed, with only 6 (1.9%) having tertiary education. The majority of households had between 5–10 members 188 (59.5%), that are predominantly Protestant 166 (36.7%) with regards to religious belief. Concerning participation in agriculture, 257 (81.3%) had access to land, and 215 (68.0%) of the respondents reported accessing less than one acre, with only 123 (38.8%) reporting use of hired land. Most of the households not only had kitchen gardens 232 (73.4%) but also reported casual labour 132 (41.8%) as a main

**Table 1. Characteristics of the participants.**

| Variables | Level | Overall (n = 316) |
|---|---|---|
| Gender | Male | 212 (67.1) |
| | Female | 104 (32.9) |
| Age categories (years) | Below 18 | 1 (0.3) |
| | 18 - 30 | 103 (32.6) |
| | 31- 45 | 146 (46.2) |
| | 46 and above | 66 (20.9) |
| Marital status | Married | 236 (74.7) |
| | Separated | 23 (7.3) |
| | Divorced | 9 (2.8) |
| | Single | 13 (4.1) |
| | Widowed | 35 (11.1) |
| Occupation | Unemployed | 157 (49.7) |
| | Employed | 11 (3.5) |
| | Small scale trading | 26 (8.2) |
| | Casual labour | 101 (32.0) |
| | Farming | 21 (6.6) |
| Education level | None | 129 (40.8) |
| | Primary | 128 (40.5) |
| | Secondary | 53 (16.8) |
| | Tertiary | 6 (1.9) |
| Household size | Small (1–4) | 116 (36.7) |
| | Medium (5–10) | 188 (59.5) |
| | Large (> 11) | 12 (3.8) |
| Religion | Catholic | 83 (26.3) |
| | Moslem | 12 (3.8) |
| | Protestant | 116 (36.7) |
| | Anglican | 36 (11.4) |
| | Seventh-day Adventist | 14 (4.4) |
| | Other | 55 (17.4) |
| Participation in agriculture | No | 24 (7.6) |
| | Yes | 292 (92.4) |
| Access to agricultural land | Yes | 257 (81.3) |
| | No | 59 (18.7) |
| Acres of agricultural land | Less than 1 acre | 215 (68.0) |
| | More than 1 acre | 42 (13.3) |
| Presence of kitchen garden | Yes | 232 (73.4) |
| | No | 84 (26.6) |
| Main source of income | Sale of crops | 20 (6.3) |
| | Sale of animals | 3 (0.9) |
| | Casual labour | 132 (41.8) |
| | Brewing alcohol | 14 (4.4) |
| | Small scale business | 56 (17.7) |
| | Handcrafts | 12 (3.8) |
| | Petty trade | 8 (2.5) |
| Source of household food | Own food production | 52 (16.5) |
| | Market purchase | 21 (6.6) |

*(Continued)*

**Table 1.** (Continued)

| Variables | Level | Overall (n=316) |
|---|---|---|
| | Gifts/donation | 243 (76.9) |
| Distance to the market (km) | Less than 1 km | 155 (49.1) |
| | More than 1 km | 161 (50.9) |
| Access to hired land | No | 193 (61.1) |
| | Yes | 123 (38.9) |
| Monthly income (UGX) | Median (IQR) | 60,000 (3,000-200,000) |
| Monthly expenditure (UGX) | Mean (SD) | 67731(62,353) |

**Note:** Values are presented as numbers (%), medians (interquartile ranges), or means±standard deviations. SD=standard deviation. IQR=Interquartile range

source of income. Overall, households earned a median income of 60,000 (interquartile ranges, IQR:=3,000–200,000) and spent on average 67,731 (standard deviation, SD=62353) Ugandan shillings (UGX) on food per month.

### Food groups consumed from own food production

Fig 1 shows the proportion of different food groups from own food production consumed. Own food production contributed greatly (85%) to the dietary intake of vegetables and moderately to the intake of pulses and nuts (65%). The percentage of respondents that consumed animal products from their own food production was very low (3% for milk and dairy products; 7.5% for meat). Surprisingly, the proportion of households that consumed eggs and fish was the same (10%) (Fig 1).

### Current household dietary diversity categories based on own food production

Table 2 presents current household dietary diversity categories based on food production among 316 households. The majority (58.5%) were in the medium dietary diversity category, followed by high diversity (32.6%) while the proportion of respondents with low dietary diversity presents at (8.9%). The overall mean dietary diversity score was 5.03 (standard deviation, SD=1.34).

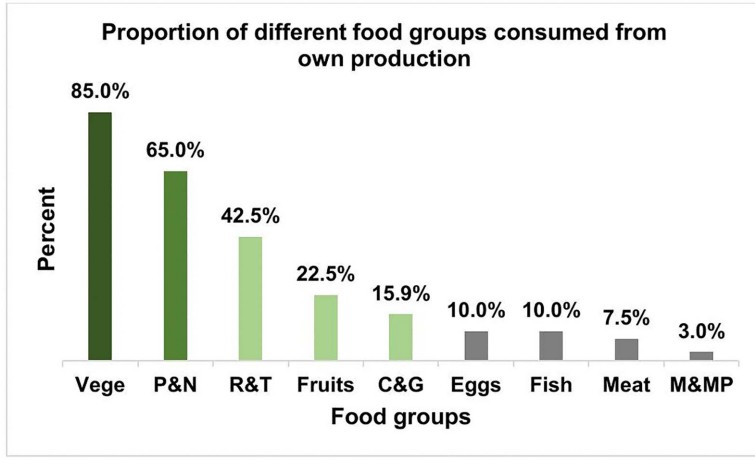

**Fig 1. Food groups consumed from own food production. Note:** Vege=Vegetables, P&N=Pulses and nuts, R&T=Roots and tubers, Fruits, C&G=Cereals and grains, Eggs, Fish, Meat, M&MP=Milk and milk products.

**Table 2. Current household dietary diversity categories based on own food production.**

| Categories of current household dietary diversity | Overall (n = 316) |
|---|---|
| Low dietary diversity (less than 3 food groups) | 28 (8.9) |
| Medium dietary diversity (4–6 food groups) | 185 (58.5) |
| High dietary diversity (more than 6 food groups) | 103 (32.6) |
| Mean (SD) | 5.03 (1.34) |

**Note:** SD = standard deviation

## Contribution of own food production to energy and selected nutrients

Table 3 shows the average annual amount of energy and specific nutrients that households require and can acquire from their own food production, as well as the average contribution of their own food production. Overall, the findings indicate that the average amount of energy and specific nutrients obtained from self-production is relatively small. However, households' average consumption of zinc (1749 mg/yr) from their own food production is quite low, whereas their average consumption of vitamin A (793247.44 µg/yr) is high. Own food production made a fair contribution to vitamin A intake (41.1%) in comparison to iron (10.2%) in terms of the relative mean contribution to energy and specific nutrients. Although the mean consumption of zinc was very low (1749 mg/yr), own food production contributed more to consumption of zinc (8.1%) compared to energy (5.1%). However, the findings show that own-produced food only contributed 2.17 percent of the households' annual calcium intake.

## Proportion of households that met recommended annual allowance from own food production

Table 4 shows the percentage of households that produced their own food and fulfilled the annual recommended allowance for energy and essential nutrients selected in the study. According to the findings, relatively few homes produced enough food to meet the annual RDAs. One noteworthy finding is that none of the families achieved the yearly RDA for calcium from own food production, whereas 19.3% of the households achieved annual requirement for vitamin A from own food production.

## Predictors of current household dietary diversity

Table 5 presents the result of Poisson regression for predictors of current household dietary diversity (HDDS). The results indicate that refugees with primary or secondary levels of education, those having less than 1 acre of agricultural land, and those with no kitchen garden are associated with reduced expected frequency of HDDS. Conversely, having access

**Table 3. Contribution of own food production to energy and selected nutrients.**

| Nutrients | Annual recommended dietary allowance | Annual Consumption from own food production | Mean contribution |
|---|---|---|---|
| | Mean (SE) | Mean (SE) | Percentage |
| Protein (g/yr) | 105050.91 (52394.619) | 9661.76 (634.176) | 9.2% |
| Energy (k/cal/yr) | 6165029.62 (3074905.174) | 316155.22 (20560.294) | 5.1% |
| Zinc (mg/yr) | 21529.24 (10738.143) | 1749.53 (120.329) | 8.1% |
| Calcium(mg/yr) | 3019022.08 (1505838.963) | 65714.94 (6765.712) | 2.2% |
| Iron (mg/yr) | 31348.55 (15634.985) | 3202.64 (234.423) | 10.2% |
| Vitamin (µg/yr) | 1928397.48 (961784.989) | 793247.44 (121530.432) | 41.1% |

**Note:** SE -Standard error.

**Table 4.** Proportion of households that met RDA from own food production.

| Energy and selected nutrient | Proportion that met RDA (%) |
|---|---|
| Energy | 1.0 |
| Calcium | 0.0 |
| Iron | 5.0 |
| Zinc | 4.8 |
| Vitamin A | 19.3 |
| Protein | 4.8 |

**Table 5.** Predictors of current household dietary diversity.

| Variables | Level | Coef (95% CI) | p-value |
|---|---|---|---|
| Gender | Male | ref | |
| | Female | 0.02 (−0.04, 0.08) | 0.583 |
| Occupation | Farming | ref | |
| | Unemployed | −0.05 (−0.15, 0.06) | 0.400 |
| | Casual labour | 0.01 (−0.08, 0.11) | 0.782 |
| | Employed | 0.01 (−0.21, 0.22) | 0.958 |
| | Small scale trading | 0.06 (−0.06, 0.17) | 0.344 |
| Education | Tertiary | ref | |
| | None | −0.16 (−0.37, 0.04) | 0.119 |
| | Primary | −0.22 (−0.42, −0.01) | 0.042* |
| | Secondary | −0.26 (−0.46, −0.05) | 0.015* |
| Participation in agriculture | No | ref | |
| | Yes | 0.09 (−0.01, 0.21) | 0.068 |
| Access to agricultural land | No | ref | |
| | Yes | 0.18 (0.08, 0.29) | <0.001*** |
| Acres of agricultural land | More than 1 acre | ref | . |
| | Less than 1 acre | −0.10 (−0.18, −0.02) | 0.013* |
| Presence of kitchen garden | Yes | ref | |
| | No | −0.09 (−0.17, −0.02) | 0.012* |
| Access to hired land | Yes | ref | |
| | No | 0.002 (−0.06, 0.07) | 0.948 |
| Monthly expenditure | | −2.92e-07 (−7.49e-07, 1.64e-07) | 0.209 |
| Monthly income | | 3.612e-07 (1.192e-07, 6.033e-07) | 0.003** |
| Mean VIF | | 1.38 | |

**Note:** 1) 95% confidence intervals in brackets; and 2) Statistical significance codes at 95% CI: *$p<0.05$, **$p<0.01$, *** $p<0.001$; 3) ref-Reference categories adopted in the analysis

to agricultural land, and for every 1Ugx increase in monthly income (range, 3,000–200,000 Shillings) is associated with increased expected frequency of HDDS.

The test for correctness of the link function indicated no significant result ($p=0.276$) to imply the log link was correctly specified. The Pearson-based dispersion value (1/degree of freedom (df) = 0.484) was significantly lower than the threshold (more than 1) that would indicate concern regarding over-dispersion. Hence, there was no evidence of substantial

over-dispersion in the data. These findings show the appropriateness of using the Poisson regression model in explaining the data in the investigations.

## Predictors of nutrient intake adequacy

Results for energy and specific nutritional components are displayed in Table 6. Nutrient intake adequacy was found to be significantly associated with household head's occupation, educational attainment, lack of a kitchen garden, and access to rented land. Energy, protein, calcium, iron, and zinc consumption are associated with a reduction of 0.169, 0.289, 0.095, 0.396, and 0.247 on average when one is unemployed. On average, households without a kitchen garden consumed 0.075, 0.121, 0.033, 0.110, and 1.219 units less energy, proteins, calcium, zinc, and vitamin A, respectively. On average, zinc intake was associated with 0.11 units higher for household heads with secondary education compared to those without any education. Additionally, on average, having access to rented land boosted intake of energy by 0.08, protein by 0.16, calcium by 0.038, iron by 0.2, and zinc by 0.157 units.

## Discussion

This study examined contribution of own food production to dietary diversity and nutrient intake adequacy, and their predictors in the Palorinya refugee settlement, Obongi district. The findings indicate that most households consumed five food groups including vegetables, pulses/nuts, roots and tubes, fruits, cereals and grains per day from own food production with vegetables contributing the greatest to their dietary intake. Approximately, six out of ten participants are in the median current household dietary diversity. This result is different from that of Abou-rizk et al. [44] who reported that Syrian refugees in Lebanon consumed on average three food groups. This difference might be due to geographical location as well as differences in physical access to different food groups. Koppmair et al. [11] posit that it might be due to the production of staples only with limited attention given to other crops. Increasing production diversity by the households in refugee settings would be ideal. The high consumption of vegetables from own food produce in this study could largely be due to ease of cultivation especially green leafy vegetables. Besides, the cultivation of vegetables requires small pieces of land and capital while providing income for poorer households. The proportion of households who consumed meat, eggs, fish, milk and milk products from their own food production was low. The results for low consumption of animal-sourced food groups from own food production are consistent with a past study [12].The significant expense and extensive space needed for the production of animal-sourced foods render the minimal contribution of local food production to these categories unsurprising. In this refugee context, capital and land—both of which are limited—are essential inputs for animal production. With regards to the annual recommended dietary allowance (RDA), the findings indicate a significantly low proportion of households that met annual RDA of micronutrients from own food produced. This finding aligns with the study of [17]. This might be due to low production of crops rich in the selected nutrients related to infertile soils, small degraded land, climatic conditions. As reported by Sakho-jimbira & Hathie [45], agriculture in sub-Saharan Africa is rain fed. Palorinya refugee settlement in recent times has been plagued by unpredictable rainfall patterns compounded with lengthy drought, unexpected floods and extreme temperatures which limits agricultural production on small, degraded plots of land thus reducing crop yield and subsequently nutrient consumption. Additionally, the majority of the households were unable to meet their zinc needs. Prior study supports low zinc intake in families involved in agricultural production [12]. This could be due to limited production and consumption of zinc rich foods. The proportion of households consuming adequate vitamin A while consuming little of their own food produce was higher than that of other selected nutrients. Increased consumption and production of greens like amaranths may be the cause of this. The simplicity of vegetable production, combined with its minimal production factor requirements, may be a plausible explanation [46]. Consuming enough vitamin, A is essential for immune building, infection prevention, and the formation of healthy skin [47]. Surprisingly, all the households did not meet the recommended daily allowance for calcium from own food production. Low production of calcium-rich crops, like simsim, and/or post-harvest losses of such crops could be the cause of

**Table 6. Predictors of energy and nutrient intake adequacy.**

| Variables | Model 1 (Energy) | Model 2 (Protein) | Model 3 (Calcium) | Model 4 (Iron) | Model 5 (Zinc) | Model 6 (Vitamin A) |
|---|---|---|---|---|---|---|
| | Coef. | Coef. | Coef. | Coef. | Coef. | Coef. |
| | (95% CI) | (95% CI) | (95% CI) | (95% CI) | (95% CI) | (95% CI) |
| **Intercept** | 0.055 | 0.051 | 0.038 | 0.059 | 0.093 | 1.365 |
| | (−0.059, 0.169) | (−0.157, 0.258) | (−0.020, 0.096) | (−0.219, 0.337) | (−0.091, 0.278) | (−0.318, 3.047) |
| **Gender** | | | | | | |
| Male | ref | | | | | |
| Female | −0.010 | −0.016 | −0.0003 | 0.012 | −0.013 | −0.206 |
| | (−0.061, 0.040) | (−0.108, 0.075) | (−0.026, 0.025) | (−0.110, 0.134) | (−0.094, 0.068) | (−0.945, 0.532) |
| **Occupation** | | | | | | |
| Not employed | ref | | | | | |
| Employed | −0.169** | −0.289** | −0.095*** | −0.396** | −0.247** | −0.712 |
| | (−0.309, −0.030) | (−0.543, −0.036) | (−0.166, −0.024) | (−0.736, −0.056) | (−0.473, −0.021) | (−2.771, 1.346) |
| Small scale trading | −0.001 | 0.0001 | −0.006 | 0.096 | −0.054 | −0.037 |
| | (−0.088, 0.085) | (−0.157, 0.158) | (−0.049, 0.038) | (−0.115, 0.307) | (−0.194, 0.087) | (−1.314, 1.240) |
| Casual labour | 0.019 | 0.044 | 0.006 | 0.018 | −0.001 | 0.360 |
| | (−0.038, 0.076) | (−0.059, 0.148) | (−0.022, 0.035) | (−0.121, 0.156) | (−0.093, 0.091) | (−0.479, 1.199) |
| Farming | −0.002 | −0.003 | 0.005 | −0.009 | −0.064 | 0.074 |
| | (−0.101, 0.097) | (−0.182, 0.177) | (−0.045, 0.055) | (−0.250, 0.232) | (−0.224, 0.096) | (−1.384, 1.532) |
| **Education level** | | | | | | |
| None | ref | | | | | |
| Primary | −0.001 | −0.0002 | −0.003 | −0.071 | 0.029 | 0.075 |
| | (−0.049, 0.047) | (−0.088, 0.088) | (−0.027, 0.022) | (−0.188, 0.047) | (−0.049, 0.107) | (−0.637, 0.787) |
| Secondary | 0.057* | .073 | 0.011 | 0.044 | 0.110** | 0.444 |
| | (−0.010, 0.123) | (−0.047, 0.193) | (−0.022, 0.045) | (−0.117, 0.206) | (0.003, 0.217) | (−0.533, 1.420) |
| Tertiary | 0.088 | 0.106 | 0.065 | 0.235 | 0.157 | 0.548 |
| | (−0.093, 0.270) | (−0.224, 0.435) | (−0.027, 0.157) | (−0.206, 0.676) | (−0.136, 0.450) | (−2.124, 3.221) |
| **Participation in agriculture** | | | | | | |
| No | ref | | | | | |
| Yes | 0.073 | 0.153* | 0.014 | 0.180 | 0.104 | 0.087 |
| | (−0.025, 0.170) | (−0.024 0.330) | (−0.035, 0.064) | (−0.057, 0.417) | (−0.054, 0.261) | (−1.348, 1.522) |
| **Access to agricultural land** | | | | | | |
| Yes | ref | | | | | |
| No | 0.006 | 0.057 | −0.025 | 0.047 | −0.008 | −0.976* |
| | (−0.066, 0.077) | (−0.073, 0.187) | (−0.061, 0.011) | (−0.127, 0.221) | (−0.124, 0.107) | (−2.030, 0.079) |
| **Presence of kitchen garden** | | | | | | |
| Yes | Ref | | | | | |
| No | −0.075** | −0.121** | −0.033** | −0.115 | −0.110** | −1.219** |
| | (−0.138, −0.012) | (−0.235, −0.006) | (−0.065, −0.001) | (−0.268, 0.038) | (−0.211, −0.008) | (−2.146, −0.292) |
| **Access to hired land** | | | | | | |
| No | ref | | | | | |
| Yes | 0.084*** | 0.161*** | 0.038*** | 0.201*** | 0.157*** | −0.016 |
| | (0.033, 0.135) | (0.069, 0.254) | (0.013, 0.064) | (0.077, 0.325) | (0.074, 0.239) | (−0.767, 0.736) |
| **Monthly expenditure** | −8.10e-8 | −8.31e-08 | 2.18e-8 | −1.66e-7 | 1.39e-9 | −7.17e-8 |

*(Continued)*

**Table 6.** (Continued)

| Variables | Model 1 (Energy) | Model 2 (Protein) | Model 3 (Calcium) | Model 4 (Iron) | Model 5 (Zinc) | Model 6 (Vitamin A) |
|---|---|---|---|---|---|---|
| | Coef. | Coef. | Coef. | Coef. | Coef. | Coef. |
| | (95% CI) | (95% CI) | (95% CI) | (95% CI) | (95% CI) | (95% CI) |
| | (−2.85e-7, 1.23e-7) | (−4.54e-7, 2.88e-7,) | (−8.18e-8, 1.25e-7) | (−6.64e-7, 3.31e-7) | (−3.29e-7, 3.32e-7) | (−3.08e-6, 2.94e-6) |
| Monthly income | 3.57e-7 | 5.55e-7 | 1.04e-7 | 8.16e-7 | 1.77e-7 | 1.66e-06 |
| | (−7.46e-8, 7.88e-7) | (−2.28e-7, 1.34e-6) | (−1.15e-7, 3.22e-7) | (−2.34e-7, 1.87e-6) | (−5.21e-7, 8.74e-7) | (−4.70e-6, 8.01e-6) |
| Mean VIF | 1.15 | | | | | |

**Note:** 1) 95% confidence intervals in brackets; and 2) Statistical significance codes at 95% CI: *p<0.05, **p<0.01, *** p<0.001; 3) ref-Reference categories adopted in the analysis

this. Additionally, the extreme inadequate intake of calcium by the households could be due to the fact that animal products were eliminated while capturing data of the previous own food produce. Moreover, animal products especially fish, milk and milk products represent one of the rich sources of calcium hence this finding may not represent clear picture of calcium intake of the households in this settlement. In terms of the predictors, this study showed that current household dietary diversity was significantly associated with the education level of the household head, access to agricultural land, presence of kitchen gardens, income, and access to hired land. Specifically, household heads reporting attendance at primary or secondary school were associated with reduced expected frequency of current HDDS compared to household heads with tertiary education. This result shows the importance of higher education in households since attaining tertiary education was associated with improved current HDDS. A previous study in Tanzania found that primary school level of participants is associated with low food scores [20]. Attaining a higher level of education may result in increased chances of formal employment hence more money for buying more diversified food groups. As reported by Ansem et al. [24], families with higher education have better nutritional quality compared to households with low education.

Furthermore, households with access to agricultural land were associated with an increased frequency of current HDDS compared to their counterparts without access to agricultural land. Similarly, higher dietary diversity scores were found among rural households participating in food production [9]. This result demonstrates the prominent role of food production among vulnerable communities in nutrition.

This study showed that households with less than one acre of agricultural land were associated with reduced frequency of current HDDS compared to those with more than one acre. This suggests that cultivation of small size of land is correlated with a low intake of food groups while large land cultivation is correlated with the intake of a high number of food groups. Additionally, land as an input of production forms an indirect pathway between diet and nutrition [48]. Increased land size may contribute to more available space to produce a variety of foods and rear animals thus improving nutrient intake and household income. The latter may be used for the purchase of diverse nutritious food groups, such as fruits and vegetables. On a similar note, increased access to land by women improves income and credit accessibility as well as household social capital, and thus the purchasing power of the household increases resulting in more access to diverse food groups [49]. A previous study reported similar results where households owning/receiving larger land were likely to produce more diverse crops in both refugee and host households [24]. This inconsistency may suggest that the impact of the size of land on current HDDS may be context-specific and that the production diversity is more important than the size of land in improving current HDDS. Reforming refugee land policy by increasing the size of land, negotiating

for reduced rental prices, and having clear-cut user documents on rental land with host communities should be advocated for by United Nations High Commissioner for Refugees (UNHCR) and human rights advocacy bodies.

We found that households with a kitchen garden are associated with an increased frequency of current HDDS compared to those with no kitchen garden. This might be due to the intake of diverse food groups from the kitchen garden. Comparable results were also reported by Taruvinga and colleagues [20] where rural households with access to home gardens were more likely to move into a high dietary diversity status.

Additionally, monthly household income had a positive and significant association with current HDDS. Increased income signifies more access to a variety of foods as a result of increased purchasing power. Additionally, this could also be due to increased disposable income on farm inputs, labour, and subsidiary technology which improves agricultural production resulting in the production of a more diverse food group. In line with this study, Parappurathu et al. [21] found that with improved income, the dietary diversity of households increases while evidence shows that, with less income, dietary diversity and calorific intake become inadequate among households [50].

In terms of predictors of nutrient intake adequacy, the results show that the household head's occupation, educational attainment, kitchen garden, and availability of rented land. On average, an unemployed person was significantly associated with less consumption of energy, protein, calcium, iron, and zinc than someone who is employed. This supports research showing that employment raises vitamin and mineral intake [51]. Formal employment is a measure of household income that can increase access and consumption of diverse nutritious food. Moreover, it can increase agricultural output and hence more nutrient consumption.

Education often has the biggest influence on dietary consumption when compared to other factors, according to studies that have examined socioeconomic factors influencing nutrient intake and included education status. For instance, Muggaga et al. [17], found that household education status significantly impacted the adequacy of nutrient intake. Their results are consistent with the results in this study, which shows that zinc intake was positively and significantly associated with educational attainment. In support of this, Rezazadeh et al. [52] found that parents who had higher levels of education also consumed more nutrients. Education may increase access to, and utilization of, the knowledge needed to boost agricultural production, in addition to raising household income through employment.

Additionally, the results of this study showed that homes without a kitchen garden were associated with lower nutritional intake. Compared to households with kitchen plots, households without a kitchen garden were on average linked to lower intakes of energy, proteins, calcium, zinc, and vitamin. These results are consistent with those of Singh et al. [53] who reported that eating more fruits and vegetables from kitchen gardens boosted intake of beta-carotene, calcium, and protein. Kitchen gardens improve nutrient intake by increasing access to diverse nutrient dense foods and improve the purchasing power of households by improving income from sale of surplus produce which can be used to purchase nutritious foods. It will be advantageous to give households the technical guidance required to establish a nutrient-dense kitchen garden and to supply them with nutrition-sensitive cultivars.

The capacity to rent more land was another factor that was significantly associated with nutrient intake adequacy. Low nutritional intake was reported by respondents who did not rent land. Low intake was also noted by Sarkar [54] among Indian households with restricted access to land. In a similar vein, a study [55] revealed that the lack of cultivable land among land evictees affected the quality of their meals. Given the refugees' modest landholdings and restricted access to land, this outcome is not shocking.

## Strengths and limitations of this study

Our study has strengths and limitations. To the best of our knowledge, this is the first study in the Palorinya refugee settlement, Obongi district, to utilize the relatively untapped potential of using regularly performed household consumption surveys (HCS) to look into the connection between dietary diversity, nutrient intake and agriculture. This survey provides precise information on current micronutrient intakes and the size of the gaps between intakes and requirements, which

can be used to potentially close these gaps with various intervention programs. Limitations include the possibility of recall bias for self-reported responses during the survey without any means of validating the responses, thus influencing the validity of the findings. Additionally, our analysis did not use a nationally representative sample, so the findings are not likely generalizable to the entire country and other similar settings. Lastly, dietary intake and own food production are subject to seasonal fluctuations, therefore, a single 24-hour recall employed in this study only provides a snapshot of current dietary diversity and does not capture habitual or seasonal dietary patterns which could influence current household dietary diversity and nutrient intake outcomes.

## Conclusion and recommendations

This study examined the contribution of own food production to current household dietary diversity and nutrient intake adequacy among refugee households in Palorinya Settlement, Uganda. The study revealed that while own food production contributed highly to vegetable consumption, it did not significantly support adequate intake of other essential food groups such as animal-sourced products, fruits, and dairy. Moreover, nutrient intake adequacy from own food production was alarmingly low across all essential micronutrients. The results confirm that vitamin and mineral deficiencies are common in this refugee settlement. Key predictors of current dietary diversity and nutrient adequacy included education level, access to agricultural land, kitchen gardening, occupation, income, and the use of hired land. These findings reinforce the multidimensional nature of food and nutrition security and underscore the need for holistic, integrated solutions.

The implications of these findings are significant for humanitarian and development actors. First, own food production while valuable for enhancing food and nutrition security should not be viewed as a standalone solution to refugee nutrition problems. It must be complemented by market access, food supplementation, and targeted nutrition education. Second, land allocation policies in refugee settlements should be revisited to enable diversified farming systems that include nutrient-dense crops and small livestock. Third, the promotion of kitchen gardens, particularly those that include fruit-bearing plants and small-scale animal husbandry, offers a practical pathway to improve nutrient intake adequacy. Future research should focus on longitudinal multiple assessments to evaluate the seasonal and long-term effects of own food production on household nutrition. There is also a need to investigate the interplay between food aid, market access, and own food production. In conclusion, while own food production plays a crucial role in improving food security among refugee households, its contribution to dietary diversity and nutrient intake adequacy is limited and uneven across the households. Addressing malnutrition in refugee settlements requires a holistic, multi-sectoral strategy that aligns agricultural, educational, economic, and health-based solutions. Only through such integration can we achieve sustainable nutritional security in fragile humanitarian contexts.

## Supporting information

**S1 File. Dataset.**
(CSV)

## Acknowledgments

We thank the research assistants for quality data collection and all the co-authors for their wonderful insights.

## Author contributions

**Conceptualization:** Martine Asobasi, Peter Omara.

**Data curation:** Martine Asobasi, Saidi Appeli, Peter Omara, Solomon Olum.

**Formal analysis:** Martine Asobasi, Peter Omara, Solomon Olum.

**Investigation:** Martine Asobasi, Peter Omara.

**Methodology:** Martine Asobasi, Saidi Appeli, Solomon Olum.

**Project administration:** Martine Asobasi.

**Software:** Martine Asobasi, Saidi Appeli.

**Supervision:** Martine Asobasi, Saidi Appeli, Solomon Olum.

**Validation:** Martine Asobasi, Saidi Appeli, Peter Omara, Solomon Olum.

**Visualization:** Martine Asobasi, Saidi Appeli, Peter Omara, Solomon Olum.

**Writing – original draft:** Martine Asobasi, Saidi Appeli, Peter Omara, Solomon Olum.

**Writing – review & editing:** Martine Asobasi, Saidi Appeli, Peter Omara, Solomon Olum.

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
