## [Decision Letter · Decision Letter 0]

26 Oct 2025

Dear Dr. Asobasi,

Thank you for submitting your manuscript to PLOS ONE. After careful consideration, we feel that it has merit but does not fully meet PLOS ONE’s publication criteria as it currently stands. Therefore, we invite you to submit a revised version of the manuscript that addresses the points raised during the review process.

We look forward to receiving your revised manuscript.

Kind regards,

Olutosin Ademola Otekunrin

Academic Editor

PLOS ONE

Journal Requirements:

3. You indicated that you had ethical approval for your study. In your Methods section, please ensure you have also stated whether you obtained consent from parents or guardians of the minors included in the study or whether the research ethics committee or IRB specifically waived the need for their consent.

4. Please provide additional information regarding the considerations made for the Refugee included in this study. For instance, please discuss whether participants were able to opt out of the study and whether individuals who did not participate receive the same treatment offered to participants.

5. We note that your Data Availability Statement is currently as follows: “All relevant data are within the manuscript and its Supporting Information files.”

6. We notice that your supplementary figures are uploaded with the file type 'Figure'. Please amend the file type to 'Supporting Information'. Please ensure that each Supporting Information file has a legend listed in the manuscript after the references list.

Additional Editor Comments:

Experts in the field have reviewed your manuscript and you are expected to address their comments as early as possible. Thank you.

Reviewer's Responses to Questions

**Comments to the Author**

1. Is the manuscript technically sound, and do the data support the conclusions?

Reviewer #1: No

Reviewer #2: Partly

2. Has the statistical analysis been performed appropriately and rigorously?

Reviewer #1: N/A

Reviewer #2: No

3. Have the authors made all data underlying the findings in their manuscript fully available?

Reviewer #1: No

Reviewer #2: Yes

4. Is the manuscript presented in an intelligible fashion and written in standard English?

Reviewer #1: Yes

Reviewer #2: Yes

Reviewer #1: Dear Authors,

you have presented an interesting study. However, your methodology is inconsistently described and not sufficiently referenced. You are reporting to have used the HDDS - based on which reference? It is also not clear why you aggregrated tht food groups to 9 food groups. Which ones were merged? Apart from this: you wrote you applied a 24h recall but then you used an FFQ? In addition you are presenting the results as yearly intake based on one 24h recall ignoring any seasonal changes in availability, production and affordability.

Reviewer #2: Nutrition security in refugees are an important and relevant topic.

A 24 hour recall on diet diversity is insufficient to determine HHD. - Suggest indicate this strongly and maybe call it current diet diversity - and include this as one of the limitations.

It might be also seasonal.

There might also be a recall bias in terms of validity - discuss reliability, plausibility and limitations in this case.

The sample only includes one settlement area - and generalisation for the whole of Uganda is thus ungrounded.

Is the statement of 0% of households relating to Calcium intake measured or assumed? Please elaborate and explain, clarify in paper.

Clarify the contribution of own production as well as nutrient intake and the adequecy thereof - also clearly make a distinction between the two.

Add VIF values - how was this calculated?

Please look at the article:

Analysing the contribution of urban agriculture towards urban household food security in informal settlement areas

JW Swanepoel, JA Van Niekerk, P Tirivanhu

Development Southern Africa 38 (5), 785-798

PLease explain unemployed.

Correct the following - casual labor table say 41.8% and text 48.1%

PLease clarify - 95% fell short - are 5% adequate? proteien RDA - since table shows 4.8% - clearly explain.

There are inconsistencies between own production/self production and household food production

Add missing values on tables

what does this mean?If published, this will include your full peer review and any attached files.). If published, this will include your full peer review and any attached files.

**Do you want your identity to be public for this peer review?** For information about this choice, including consent withdrawal, please see our For information about this choice, including consent withdrawal, please see our Privacy Policy .

Reviewer #1: No

Reviewer #2: **Yes:** Proj Jan Willem SwanepoelProj Jan Willem Swanepoel

---

## [Author Response · Author response to Decision Letter 1]

3 Nov 2025

We appreciate you for the feedback you provided. We have addressed all the comments.

---

## [Decision Letter · Decision Letter 1]

10 Mar 2026

Contribution of own food production to household dietary diversity and nutrient intake adequacy among refugee households in Palorinya, northwestern Uganda

PONE-D-25-39697R1

Dear Dr. Asobasi,

We’re pleased to inform you that your manuscript has been judged scientifically suitable for publication and will be formally accepted for publication once it meets all outstanding technical requirements.

Kind regards,

Olutosin Ademola Otekunrin

Academic Editor

PLOS One

Additional Editor Comments (optional):

Reviewers' comments:

Reviewer's Responses to Questions

**Comments to the Author**

Reviewer #2: All comments have been addressed

2. Is the manuscript technically sound, and do the data support the conclusions?

Reviewer #2: Yes

3. Has the statistical analysis been performed appropriately and rigorously?

Reviewer #2: Yes

4. Have the authors made all data underlying the findings in their manuscript fully available?

Reviewer #2: Yes

5. Is the manuscript presented in an intelligible fashion and written in standard English?

Reviewer #2: Yes

Reviewer #2: None at this stage. Terminology has been standarized.

Inconsistencies concerning numbers have been corrected.

Methodology have been updated and corrected.

what does this mean?If published, this will include your full peer review and any attached files.). If published, this will include your full peer review and any attached files.

**Do you want your identity to be public for this peer review?** For information about this choice, including consent withdrawal, please see our For information about this choice, including consent withdrawal, please see our Privacy Policy .

Reviewer #2: **Yes:** Prof Jan Willem SwanepoelProf Jan Willem Swanepoel

---

## [Editor Report · Acceptance letter]

PONE-D-25-39697R1

PLOS One

Dear Dr. Asobasi,

I'm pleased to inform you that your manuscript has been deemed suitable for publication in PLOS One. Congratulations! Your manuscript is now being handed over to our production team.

Kind regards,

on behalf of

Dr. Olutosin Ademola Otekunrin

Academic Editor

PLOS One